# Indeterminacy of the Diagnosis of Sudden Infant Death Syndrome Leading to Problems with the Validity of Data

**DOI:** 10.3390/diagnostics12071512

**Published:** 2022-06-21

**Authors:** Ivana Olecká, Martin Dobiáš, Adéla Lemrová, Kateřina Ivanová, Tomáš Fürst, Jan Krajsa, Petr Handlos

**Affiliations:** 1Department of Christian Social Work, Sts Cyril and Methodius Faculty of Theology, Palacký University Olomouc, 779 00 Olomouc, Czech Republic; ivana.olecka@upol.cz; 2Department of Forensic Medicine and Medical Law, Faculty of Medicine and Dentistry, Palacký University Olomouc, 779 00 Olomouc, Czech Republic; 3Department of Public Health, Faculty of Medicine and Dentistry, Palacký University Olomouc, 779 00 Olomouc, Czech Republic; adela.lemrova01@upol.cz (A.L.); katerina.ivanova@upol.cz (K.I.); 4Department of Mathematical Analysis and Applications of Mathematics, Faculty of Science, Palacký University Olomouc, 779 00 Olomouc, Czech Republic; tomas.furst@seznam.cz; 5Department of Forensic Medicine, Faculty of Medicine, Masaryk University & St. Anne’s University Hospital Brno, 602 00 Brno, Czech Republic; jan.krajsa@fnusa.cz; 6Institute of Pathology, Faculty of Medicine, University of Ostrava & University Hospital in Ostrava, 701 03 Ostrava, Czech Republic; petr.handlos@seznam.cz

**Keywords:** death, infant, infection, injury, sudden, SIDS, suffocations, unexpected, validity of data, violent

## Abstract

The validity of infant mortality data is essential in assessing health care quality and in the setting of preventive measures. This study explores different diagnostic procedures used to determine the cause of death across forensic settings and thus the issue of the reduced validity of data. All records from three forensic medical departments that conducted autopsies on children aged 12 months or younger (n = 204) who died during the years 2007–2016 in Moravia were included. Differences in diagnostic procedures were found to be statistically significant. Each department works with a different set of risk factors and places different emphasis on different types of examination. The most significant differences could be observed in sudden infant death syndrome and suffocation diagnosis frequency. The validity of statistical data on the causes of infant mortality is thus significantly reduced. Therefore, the possibilities of public health and social policy interventions toward preventing sudden and unexpected infant death are extraordinarily complicated by this lack of data validity.

## 1. Introduction

Infant mortality is associated with several different biological factors—child and maternal health, pregnancy, and childbirth; as well as social factors—among which quality of and access to medical care, socio-economic conditions of society, especially that of the child’s family, health literacy of the mother, family lifestyle, maternal family role-models, indifference of institutions and the social environment towards the child [1]. Although infant mortality has decreased in most Western European countries over the last 15 years, its incidence is still perceived as problematic. Moreover, it should be noted that a child’s chances of surviving the first year of life differ ubiquitously, and even in the wealthiest countries, significant inequalities between different social groups persist. However, a considerable decrease in child mortality recorded in statistical overviews reflects the general improvement in professional mothers and childcare, especially during prenatal period [2,3].

The causes of child mortality are detailed in the statistics of EUROSTAT in the tenth revision of the International Statistical Classification of Diseases and Related Health Problems (ICD-10) according to the World Health Organization [4]. Perinatal conditions (P00–P96) present the most frequent cause of neonatal mortality, followed by congenital malformations, deformations, and chromosomal abnormalities (Q00–Q99). The third most frequent diagnosis are symptoms, signs, and abnormal clinical and laboratory findings not classified elsewhere (R00–R99), including sudden infant death syndrome (SIDS). The fourth most frequent causes include the external extraneous causes of mortality (Chapters XIX–XX), specifically W79 (inhalation or ingestion of food causing obstruction of the respiratory tract).

In terms of diagnosing causes of death, SIDS is the most widely discussed diagnosis. [5,6,7,8]. SIDS is a complex nosological unit (denoted by the alphanumeric code R95 in ICD), which currently cannot be explained by diagnostic criteria [9,10,11]. The autopsy findings in SIDS are largely non-specific—it is related to an infant without any sign of dysmorphism, hidden disease, or injury. However, findings related to resuscitation are possible, as well as signs of oronasal secretion [12]. Many factors involved in SIDS pathogenesis may lead to sudden death when combined together at a critical period in a child’s development [13].

Unfortunately, there is no consistency in the diagnostic approach at an international level either. Although several experts have been engaged in the development of various recommendations and the construction of questionnaires designed to collect data on the causes and circumstances of child deaths [14,15,16,17,18,19], their recommendations have not been thoroughly respected.

## 2. Literature Review

The Pubmed database was searched. The search of the texts was based on a combination of keywords (infant AND SIDS AND misdiagnos *): 44 texts were found in total. After implication of the exclusion criteria (it was not a primary study, the texts did not focus on misdiagnostics, and the articles were published before 2000), no texts were found. None of the identified texts were based on European data. Outside the European context, eight studies were identified, two of which were from Brazil, two from the USA, and the rest from Japan, India, Canada, and New Zealand. The analyzed texts all pointed out the problems in diagnostics, which can have consequences related to the non-detection of violent causes of death [6,20,21,22,23,24,25,26].

The following keywords were also searched with the following results: (SIDS) AND (wrong diagnos *)—2 texts, both irrelevant;(SIDS) AND (fals * diagnos *)—6 texts, one relevant [27];(SIDS) AND (true diagnos *)—13 texts, four relevant [28,29,30,31];(SIDS) AND (suffocation) AND (diagnos *)—47 texts, ten relevant [27,28,32,33,34,35,36,37,38,39].

(SIDS) AND (diagnos *) and (inconsistent)—11 texts, four relevant [27,34,40,41] A broad spectrum of information should be taken to ensure the reliability of SIDS diagnostics, including the child’s disease history, exact circumstances of death, and the child and mother’s social history [11,20,42,43]. Critical analyses of current medical records, including an autopsy report and other information about the suspicious death of a child (i.e., sudden, unexpected, or violent), are essential [27]. The results of the analyses provide an opportunity for a multidisciplinary discussion among the specialists involved, who can determine whether the defined stress factors (injuries, intoxications, or illnesses) played a significant role or only contributed to the child’s death [7,26,33,44].

The search strategy included a search for existing clinical guidelines. A lack of international consensus on these practices was found. As part of the conceptualization of the background to this research, we sent out a query to forensic medical centers asking about the existence of binding practices, at least at a regional level in Europe. It was found that only Italy has a standardized procedure for the diagnosis of SIDS.

Academic writings propose hundreds of etiopathogenetic theories from extended apneic episodes, allergic reactions, and immaturity of brain centers of regulation to hypoperfusion of the brainstem and cardiac conduction disorders. Hypoxia, disorders of the autonomic nervous system, effect of infection, inherited metabolism disorder, immunological abnormalities, and several external influences are most commonly cited in consideration of the SIDS causes [42]. Essential pathological–anatomical findings in typical SIDS cases are eutrophic infant and signs of asphyxia in autopsy. The most common microscopic findings are non-specific inflammatory changes in the respiratory tract and lungs (so-called atypical pneumonia) and stomach contents present in the respiratory tract. In 1993 [45], the “fatal triangle” model was developed, according to which SIDS occurs when three unfavorable conditions come together:A vulnerable stage of CNS and immune system development (vulnerable period);Predisposing factors include genetic factors (vulnerable individual);Trigger events, e.g., the child’s sleeping position, a smoking mother, or infection (vulnerable environment).

The internal findings in SIDS correspond to more or less general signs of asphyxia [43]. They may include the following: focal or diffuse pulmonary edema, the presence of areas of collapsed lung and emphysema, presence of petechiae under serous membranes of the intrathoracic organs, presence of edematous fluids in the lower respiratory tract, hypertrophy of the mesenteric lymph nodes, splenic anemia, empty bladder, and brain swelling. There is an extremely thin line between SIDS and fatal child abuse, and its clarity is further reduced by the fragmentation of diagnostic procedures [46]. Unfortunately, even a well-performed autopsy cannot always assume the role of a clear arbiter without knowledge of the circumstances of a child’s death [42]. Thus, infants eventually diagnosed with SIDS may include those whose deaths were caused by homicide or accident [43]. 

There is no definitive diagnosis of SIDS, and the underlying cause is not clear [8,47]. SIDS is a diagnosis of exclusion—as such, it is an unexpected death in terms of a previous disease, not explained by a detailed and thorough autopsy or the results of laboratory tests and accompanied by knowledge of all the circumstances from the site of death [48]. In practice, misdiagnosis may occur in cases that were actually intentional acts of violence [20,21,46,49], as it might be very difficult to distinguish the diagnosis of SIDS from that of asphyxiation (from violent or natural causes) [50]. Suffocation (T17.9; T59; T71; R09; T75.1) occurs from a lack of oxygen or accumulation of carbon dioxide in the blood. Suffocation from natural causes may occur because of morbid changes in lung tissue (e.g., infection) or lung disorders, as suspected in SIDS. Suffocation from causes other than natural may occur in several forms, such as suffocation by covering the nose and mouth, confinement in a small space, or suffocation by inhalation of gastric contents or by fluid entering the lungs [42].

According to Meadow, one mechanism in asphyctic asphyxiation is “smothering”, where the perpetrator uses his hands, other body parts or clothing, blankets, or pillows to cause mechanical obstruction of the child’s airway [43]. “Smothering” is a form of child abuse and a possible cause of childhood sudden, unexpected, or violent death. According to Meadow, airway obstruction is most common in children under one year of age. The perpetrator is most often the child’s biological mother [12]. Meadow’s report sparked a worldwide debate on whether the cause of sudden unexpected infant death (SUID) can be attributed to SIDS or suffocation [22]. The cause of death in these cases may not be clearly conclusive by an autopsy, and “smothering” may be labeled mistakenly as SIDS [20,23,46,49].

The study’s main objective and its basic hypotheses and premises were formed based on literature research. This study explored different diagnostic procedures used to determine the cause of sudden infant death syndrome across forensic settings. The main question was: What specific differences exist in the number of causes of death in children between the clinics? The following partial research questions were derived from the main research questions:
-What specific differences exist between departments regarding the diagnosed causes of death? -What specific differences exist between departments regarding how information is collected on social risk factors? -What specific differences exist between departments regarding the identification of social factors? 

## 3. Materials and Methods

### 3.1. Sample

In the Czech Republic, experts in the field of forensic medicine perform two types of autopsies according to the law: Medical autopsies aimed at determining the cause of death and clarifying all the circumstances of people who died as a result of a sudden, unexpected, or violent death;Forensic autopsies are performed when it is suspected that the death caused by a criminal offense was stated.

In the deaths of infants under one year of age, it is compulsory to perform an autopsy in the Czech Republic. We applied a retrospective content analysis of autopsy records of both medical and forensic autopsies of infants who died during a 10-year period. The use of retrospective analysis was necessary as long as autopsy files of currently open cases were not available for analysis. The evaluation involved a complete diagnostic process, including analyzing the physicians’ type and quantity of documents in the file. Thus, a prospective study would be at risk of biased results. 

The study data were collected from three departments of forensic medicine (OV, OL, and BR). All three departments together covered one geographical unit (Morava). The analyzed documents comprised all the autopsy files at the time of collection (April 2016–June 2018) that forensic medical examiners regarded as complete. All records (Table 1) from three forensic medical departments that conducted autopsies on children aged 12 months or younger (n = 204).

The OL dataset contained 43 autopsy reports on infants, OV 66 reports, and BR 95 reports. The cohort consisted of medical and forensic autopsies (n = 204) from all three departments where the investigation into the cause of death was concluded. None of the 204 cases were excluded, and therefore, the cohort included all the cases from all three departments. Annually, these three departments perform a third of all (children and adults) autopsies in the Czech Republic (n = 5000/per year).

The research sample was constructed on the Czech population, which is methodologically suitable for studying the causes of sudden and unexpected deaths of children under one year of age, whereas in the communities with extremely low infant mortality, a significant percentage of deaths caused by unavailable or low-quality health care can be ruled out. 

### 3.2. Measures

The data collected are part of a large study aimed at investigating the social determinants of death in children under one year of age who died suddenly, unexpectedly, or violently and the health literacy of their mothers [51,52]. The basis of these findings was a detailed information matrix. Everything that could be gleaned from court records was recorded for each child (and their mother), including these children’s diagnoses leading to death (i.e., causes of death) [49].

The causes and mechanisms of death based on ICD-10 diagnostics are grouped into the following six categories for statistical analysis: Childbirth-related deaths (P00–P96—certain conditions originating in the perinatal period);Injuries (burns, injuries, accidents, traffic accidents, i.e., mainly V01–Y98—external causes of morbidity and mortality and S00–T98—injury, poisoning, and other inevitable consequences of external causes, willful harm);Suffocation (W00–X59—other external causes of accidental injury and T17—foreign body in the respiratory tract (most often, cases of aspiration of milk/vomit or death due to mechanical pressure on the rib cage or covering of the nose and mouth. Five out of seventeen such cases were closed as R96–R99—ill-defined and unknown causes of mortality, with a non-specific autopsy finding suggesting an indication of some form of suffocation));Congenital malformations (Q00–Q99—congenital malformations, deformities, and chromosomal abnormalities. Most frequently, these were heart and digestive system disorders);Infections (most commonly J00–J99—respiratory and N00–N99—genitourinary diseases);SIDS R95.

This categorization appropriately classifies the study cohort regarding the most frequent causes of death at this age. It also corresponds to the forensic medicine perspective since one of the most problematic issues in forensic practice is to differentiate SIDS from asphyxiation. Group 6 includes SIDS as a group of deaths with an unknown cause, which is practically indistinguishable from the suffocation group, but this classification continues to be used.

### 3.3. Statistical Analyses

The variables were operationalized based on the death certificate (the medical opinion regarding the cause of death based on the available information at the time of death) and taking into account the theory behind the causes of death in cases of SIDS [9,10,11,20]. The frequency of each of the observed characteristics was recorded in a pre-structured sheet, which was validated in a pilot study [49]. 

Standard methods of descriptive statistics were used. The relationship between two categorical variables was analyzed by means of contingency tables and asymptotic chi-square tests of homogeneity. If the sample size was inadequate for the asymptotic test, the exact Fisher’s (factorial) test was used. The level of significance in all the tests was set to 0.05. The data were analyzed in the MatLab R2016 environment.

## 4. Results

The data from the three forensic medical departments contained 204 files, 68% of which were forensic and 32% medical autopsies. The highest proportion of autopsies was performed at the Brno department (47%) and the lowest at the Olomouc department (21%). The differences in the absolute number of autopsies at individual departments are mostly due to the size of the region.

The most obvious is the high number of suffocations diagnosed in Olomouc in contrast to zero in Ostrava. In the case of SIDS diagnosis, the ratio in Olomouc and Ostrava is reversed, albeit without a zero number of cases in Olomouc.

In Brno (BR), the causes of death are quite evenly spread; in Olomouc (OL), suffocation prevails, which is completely absent in Ostrava (OS). The departments differ mainly in the diagnoses of suffocation (OS 0%, BR 16%, and OL 40%) and SIDS (OS 23%, BR 16%, and OL 5%) (Figure 1 and Figure 2). The results confirm the differences between departments regarding the diagnosed causes of death.

If we focus on a single diagnosis—e.g., SIDS—we may produce the 3 × 2 contingency table by computing how many cases in each center died from SIDS and how many did not die from SIDS. A homogeneity test can be performed, testing the null hypothesis that the proportion of deaths from SIDS in all the centers is the same. The Freeman–Halton extension of the Fisher exact probability test was used in this case to account for cells with small entries. Homogeneity was rejected for SIDS (*p* = 0.03), suffocation (*p* < 0.001), injury (*p* = 0.016), and CDD (*p* = 0.006). On the other hand, homogeneity was not rejected for infections (*p* = 0.70), and perinatal causes (*p* = 0.73). This suggests that SIDS, suffocation, and injury may be understood differently in the three centers.

Infections and congenital developmental defects (CDD) were the most frequently diagnosed causes of death in the available records (in total, accounting for 41.6%). The third most frequent group of causes of death (16.2%) were deaths related to childbirth, followed by deaths diagnosed as SIDS (15.7%) and suffocation from various causes (15.7%). The least frequent cause of death in the study cohort is injury, accounting only for 10.8% of the cases. It should be noted that this category also included causes of death by willful harm (i.e., various blunt head injuries).

The evolution of the breakdown of the causes of death among the three centers does not show any dramatic changes (see Figure 2). The years 2012–2014 seem to be anomalous in Brno. The most dramatic effect in Figure 2 is the absence of diagnoses of suffocation and injury in Ostrava, compared to Olomouc and Brno. No test was performed due to the small number of cases in each category.

The number of infant deaths is the highest immediately after birth and between the first and fifth months of life (see Figure 3). Although the structure of infant mortality in the sample at first glance does not differ from the structure of mortality in the Czech Republic [53], further analysis reveals interesting details. While causes related to pregnancy and childbirth (perinatal deaths) can be confirmed from our study population to be consistent with national statistics, diagnoses of SIDS, suffocation, and infection were more common between the first and fifth month of life compared to national statistics.

The distribution of risk factors is shown in Table 2. All the available indicators from the documentation are combined into three indices: hostile behavior of the mother (host), adverse socio-economic factors (soc), and adverse factors related to health (health). The last column of Table 2 shows the number of cases where relevant documentation was missing. For example, the ratio of 13:2 in the first row means that in 13 cases of SIDS diagnosis in Ostrava, the documentation was missing, while in 2 cases, it was available. In each cell, the x:y score shows the number of cases where at least one of the risk factors from that category was present (x) and in how many cases all of these risk factors were absent (y). For example, the score 5:4 in the “infection row” and “health” column in Ostrava means the following. In all cases of “infection diagnosis”, in five cases, the documentation mentions adverse factors related to health, and in four cases, it does not mention them. Thus, adverse factors related to health are red-coded because they represent a risk factor for the “infection” diagnosis. Red boxes indicate an x:y ratio greater than the average of these ratios in the column for the particular department. Thus, the red boxes indicate that the factors in the respective category (host, soc, or health) are over-represented in this diagnosis (at that department). On the other hand, the green-colored boxes indicate factors that are under-represented in the diagnosis. Uncolored boxes either roughly correspond to the expected frequency ratios or the number of cases is too low to make any conclusions. 

The following categories were analyzed:
Hostile behavior of mothers (guest)—included cases not only of proven homicides but also cases of proven violent behavior with the child victim, leading to medical complications of the child leading to his death.Socio-economic factors (soc)—included aspects related to the socio-economic background of the child’s family, e.g., inadequate housing conditions and low level of hygiene in the household.Factors related to health (health)—included aspects related to health, such as poor hygiene of the child’s body, signs of poor nutrition, untreated inflammation, congenital developmental defects, etc.Relevant documentation (absent)—the absence of documents on the file containing information on the above, whether police reports, medical records, or social services information.

These results confirm the differences between departments regarding how information is collected on social risk factors.

Table 2 uncovers differences in how information on risk factors is collected. To interpret these findings, it has to be admitted that each department has different diagnostic procedures in place, which is reflected in the data collection protocol. Each department thus works with a different set of risk factors. The most significant differences can be seen in the diagnosis of SIDS and CDD. At the Olomouc department, the number of SIDS cases does not differ from the average of other diagnoses. Similarly, none of the risk factors exceeded the average score. 

However, the necessary SIDS documentation is missing in both Brno and Ostrava. In Ostrava, therefore, a significantly higher average is seen only for health-related risk factors, while the expected impact of social factors was not observed. Furthermore, social factors were not significantly higher in any of the departments. This means that differences between departments regarding identifying social factors were not confirmed. Contrary to expectations, in the case of hostile behavior, the effect of this risk factor was below average in Ostrava and Brno. In Brno, reversely, the documentation in cases of suffocation and injuries was the most detailed. For suffocation, social and health-related factors occur above the average risk factors in Brno and Olomouc. For the diagnosis of suffocation, it is evident that more emphasis was placed on the collection of social documentation in Brno and Olomouc (OL 6:6; BR 6:4), while in Brno, the risk of hospitalization was documented for suffocation (9:1). Similarly, it is worth noting in Brno the collection of documentation for the diagnosis of injury: for hostility risk (8:5) and social risk (6:7). Hostile behavior also scores above the average risk factors in the Brno department. There was no diagnosis of suffocation in Ostrava. The presented results clearly show that the data between individual centers cannot be unambiguously compared due to the absence of detailed documentation. The interpretation of data was also hampered by inconsistent diagnostics that give rise to individual categories of diagnoses, including different types of deaths. Therefore, there is a different distribution of diagnoses in each department. However, the regions are not significantly different (in terms of population health, wages, poverty, education, and other socio-economic factors). The results of the data analysis confirm the premise that the most likely reason for the observed differences in the distribution of diagnoses between departments is the variation in the extent and methodology for collecting data surrounding the infant’s death that is necessary for diagnosis and variation in diagnosis protocols.

## 5. Discussion

At the start of our research, we assume the different frequencies of SIDS, suffocation, and infection diagnoses at the three forensic medical clinics. The crucial questions are: If the differences between clinics are significant, can the data on the causes of death of children be considered valid? Could the low validity of data reported in court files indicate a bias in the extent of violence against these children? We studied the different diagnostic procedures used to state the SIDS diagnosis. The results of our study draw attention to the existing differences in diagnostic procedures among individual forensic facilities. The differences cannot be explained by the different composition of the population nor by the socio-economic differences [53,54]. Therefore, it can be assumed that deaths with non-specific signs of asphyxiation can be classified differently in various departments, sometimes as SIDS and elsewhere as suffocation (see Figure 1). This situation has persisted worldwide for a long time.

Due to the lack of unified diagnostic procedures, as shown by Sheehan et al. [8], especially in the case of SIDS, the expected data validity is reduced. This is a fact that has also been highlighted repeatedly by numerous experts [8,11,17,19]. 

Our data show that the non-uniqueness of the diagnosis starts with the information collection system. Table 2 shows that documents for social and host diagnosis were surprisingly collected for diagnoses where the diagnostic procedure does not formally require this collection (infection, injury, asphyxiation). Conversely, documentation is often absent for SIDS diagnoses where the international recommended practice emphasizes the collection of this information [49]. The consequence of missing information is that social risk factors were not recorded in these cases.

The problem is not to be found in the methods of diagnosis. However, it is apparent that differences between the forensic departments are so significant that there is no harmonized procedure for diagnosis. This assumption is supported by the fact that the reliability of diagnostic data is particularly low for diagnoses with incomplete guidelines—especially for SIDS [18]. When interpreting these data, it is essential to consider the well-known fact that, in practice, there might be some misdiagnosed cases that are deliberate violent acts and not SIDS [5,6,20,55,56]. The record of deaths diagnosed with SIDS in children older than one year is particularly striking. Eurostat data also clearly show the persistent differences in the diagnosis of SIDS among EU countries [57]. Imposing is the record of deaths diagnosed as SIDS in children older than one year. For example, since 2007, SIDS has been diagnosed in children older than one year 61 times (27 times in the United Kingdom, 13 times in Ireland, 10 times in Germany, 7 times in Belgium, and six times in Czech Republic) (Table 3). Therefore, it is evident that the problem of differential diagnosis is a local and, moreover, global problem.

In forensics practice in all high-income countries, SIDS currently represents the critical group in the mortality rate of 1–12-month old infants [42]. It is also an exceedingly problematic issue since differentiating SIDS from asphyxiation is quite difficult. Some risk factors of SIDS overlap with the risk factors of fatal maltreatment and neglect of children. It is estimated that the unacknowledged murder among these cases represents between 2 and 10% [20,58], which consequently results in undervalued data regarding the frequency of murder. Thus, the association of socio-demographic characteristics and the diagnosed causes of death of infants who died suddenly, unexpectedly, and violently was performed. 

The results shown in Figure 3 confirm the theory of Meadow [5], which described as one of the essential characteristics that children under one year die of SIDS mainly between 2 and 6 months. They most often die of suffocation by four months of age. Although Meadow examined the diagnosis of SIDS in hundreds of American children during the late 20th century, the children who died in our cohort in the 21st century continue to replicate his findings. At the same time, this fact repeatedly highlights the ambiguous SIDS diagnoses that are still being made even in children older than one year. Although the structure of infant mortality in the sample at first glance did not differ significantly from the structure of mortality in the Czech Republic [53,54], an analysis revealed exciting details. Meadow also confirms these details in his research. The slight differences in the national statistics are present due to the fact that this study involves a set of children with suspicious deaths dissected in forensic departments.

The Czech Republic is divided into 14 regions, and the differences between these regions in infant mortality are quite readily apparent. Differences in infant mortality are readily seen, especially in mortality between the genders Figure 4 and Figure 5. A statistically significant difference in the data presented in Graphs 1a and b is apparent only between the Ústí nad Labem Region, Prague, Central Bohemia, and the Vysočina Region. Tests of the differences between individual regions and the national average were performed together for all years (2008–2017) using an asymptotic chi-square test. While the Ústí nad Labem Region (*p* < 0.001) and the Karlovy Vary Region (*p* = 0.003) have a significantly increased infant mortality for boys, Prague (*p* < 0.001), Central Bohemia (*p* = 0.01) and Vysočina (*p* = 0.14) have a significantly reduced mortality. The Zlín region (*p* = 0.07) shows a value bordering on the significance level of the test, pointing to increased infant mortality in boys. Other regions do not differ significantly from the national average.

If we perform the same tests for girls, we find that the Ústí nad Labem Region has a significantly increased infant mortality (*p* < 0.001) along with the Karlovy Vary Region (*p* = 0.009). Only Prague has a significantly reduced infant mortality for girls (*p* = 0.002); the rest do not differ from the national average. The result depends on the strength of the test, and so, as fewer girl infants die, the tests are not strong enough to find a significant effect.

Comparing regions shows that the Ústí nad Labem and Karlovy Vary regions can be considered problematic regarding infant mortality. On the other hand, Prague shows better than average results, confirming the presumption of better living standards and high-quality health care.

The diagnostic differences regarding child death have multiple causes. The reason is that the information on death’s circumstances is not being collected systematically, and there is no clear guideline to establish a harmonized procedure [18,59]. There are difficulties related to classification [27], and the diagnosis of SIDS is something also a container concept; several causes could be classified as SIDS.

The absence of an international consensus on the definition of SIDS contributes to making any global comparison difficult. Nevertheless, many experts are involved in developing various recommendations and constructing questionnaires designed to collect data on the causes of and circumstances surrounding a child’s death [8,14,15,16,17,18]. The Center for Disease Control and Prevention has been developing a standardized form and guideline since 1996 [60] (Sudden, Unexplained Infant Death Investigation Guidelines). The eight-page SUIDIRF30 statement form standardizes data collection on the circumstances of death prior to autopsy. It collects the following data: identifying information, infant history, infant dietary information, pregnancy history, a record of a witness interview, details of the investigation of events, a summary of the investigation, diagrams showing the site of findings and physical injuries, and a summary for the pathologist [60]. However, these forms are not standardly used and appear to be necessary in accordance with suggestions from numerous experts [8,17,18,58,60,61] to establish a standardized binding protocol for collecting data on all circumstances that may have contributed to the child’s death, as it is not possible to establish a diagnosis of SIDS without a thorough investigation of the death and/or incident and a review of the clinical history showing no other cause of death [46]. It would also be advisable to find consensus within professional circles to define the diagnosis of SIDS clearly and to establish a diagnostic guideline [8,26,46].

A recommended procedure for pediatricians in cases of sudden infant death is currently being developed in the Czech Republic [42] to improve the diagnostics over a broad spectrum of unexpected infant deaths. Our study clearly showed that the quality of the underlying material for collecting statistical data is inadequate. The death certificate is the fundamental administrative document that serves as a basis for generating statistics on mortality. In our study cohort, we encountered inaccurately or incompletely filled death certificates. Not only in the Czech Republic but also in neighboring countries, there are no mandatory rules for the cooperation of the authorities involved—forensic practitioners, police, social workers—to meaningfully collect information about the circumstances of the death and what preceded it. The current practice is that expert groups’ recommendations [60], but their compliance is not enforced and controlled.

The higher the number of social risk factors surrounding the death, the more accurate the death certificate needs to be so that forensic examiners have all the details necessary for a precise diagnosis. Moreover, contributing to the low quality are persistent deficiencies in reporting by pediatricians in cases of suspected violence against children, lack of information, procedural coordination, and interdisciplinary cooperation among the various experts and NGOs involved [62]. Only the introduction and consistent application of standardized procedures for diagnosis and investigation of the place and circumstances of a child’s death is able to eradicate these differences.

These inadequacies lead to another problem: without the existence of guidelines, and given the variable quality of information input, the diagnostic approaches may differ—as shown by SIDS diagnostics, suffocation, infections, and injuries (Figure 1 and Figure 2). The quality of the input information for the production of statistical data thus appears to be insufficient. The reduced validity and reliability of diagnostic data then result in low validity and reliability of statistical data. If the violent death of children and quantified more realistically, the willingness of the general public to report possible suspicions of domestic violence could also increase. It can be reasonably assumed that the lack of interest in the social environment is closely related to the low level of public awareness of these cases. It would thus be easier to target the preventive measures.

### 5.1. Strengths and Limitations

The research sample was constructed on the Czech population, which is methodologically suitable for studying the causes of sudden and unexpected deaths of children under one year of age; in this population with extremely low infant mortality, a significant percentage of deaths caused by unavailable or low-quality health care can be ruled out. As a result, the authors declare that human participants’ names and other HIPAA identifiers are not included in the manuscript’s text. The relevant guidelines and regulations performed all methods. The informed consent form was made up according to the legislative requirements effective in the Czech Republic. For data analysis, extracts and copies were made from the source files only. The researchers vowed not to disclose any data that could harm specific individuals. All material obtained was to be used solely for project work with scientific, educational, and dissemination objectives. The researchers were acquainted with the obligation to conduct their research only with the knowledge of the project’s principal investigators and the need to analyze documents only in forensic medicine facilities. None of the source files or any portion of them were taken outside the relevant department. 

The research worked exclusively with documents; none of the participants were contacted by the researchers. Nevertheless, the data collected by the researchers represented sensitive personal information. It was necessary to ensure that the research outputs would not specifically identify the persons referred to in the research. Records made for data analysis are kept secure and will be shredded when those research projects that are still in progress are completed.

An important limitation of the research is the reach of the study cohort, being limited to only three departments in the Czech Republic. It may seem that this paper only presents local data in the Czech Republic, but the problem interferes on a global level, as evidenced by the literature cited above. Even the results over a 10-year time period confirm the trends of inaccurate post-mortem diagnosis of infant deaths in line with Meadow’s research. It can serve as a basis for further research using a more extensive data set.

The relevant institution should be systematically “pushed” from the bottom by experts. We intend to stir up the world’s broad expert discussion. The scientists should describe the situation and consider the influence of relevant factors. We believe this is their mission and their social responsibility. The facts need to be analyzed, and that knowledge can be transferred to the relevant bodies. It can occur after scientific discussion, and we believe our results are capable of encouraging such a discussion. Proper guidelines are needed, although they must be built on research. It should also be mentioned that the Czech environment is quite specific, as infant mortality is very low, and it is, therefore, possible to obtain data on questionable deaths. We consider this fact to be a significant advantage of the study. Although two of the three forensic departments analyzed have a shared history, the diagnostic procedures are different. 

### 5.2. Implication

According to Health for All up to the 21st Century [63], there is a lack of mutual information and coordination between various experts and non-governmental organizations to implement interdisciplinary cooperation. Twelve years later, the new WHO Health 2020 program [64] again states that no comprehensive data on violence against children in the Czech Republic are available. There are still shortcomings in child maltreatment, abuse, and neglect in the interdisciplinary cooperation of doctors, psychologists, the police, the social and legal protection department for children, etc.

When preparing the prevention programs, it is necessary to consider that when trying to decrease the infant death rate, what we can affect the most are the external causes that lead to the infants’ sudden death and result from socio-pathological phenomena. It is essential to have valid and reliable data, including statistical data, to identify the relevant prevention procedures and the accurate and corresponding structure of the infant death rate.

The relevant institutions should be systematically “pushed” from the bottom by experts. This text intends to stir up a world expert discussion. The scientists should describe the situation and consider the influence of relevant factors. The facts need to be analyzed, and then the knowledge can be transferred to the relevant bodies, which can occur after scientific discussion.

## 6. Conclusions

The low number of sudden and unexpected deaths in line with the low level of infant mortality (per 10-year period, approximately 2.3–3‰ [53] in the Czech Republic is actually good news. The availability of quality health care in the Czech Republic is a guarantee that only in unavoidable cases do children under 12 months of age die of infectious diseases and congenital defects [53,65]. A study of the deaths of infants under 12 months old in a target population with extremely low infant mortality was interesting since we could exclude deaths due to inaccessible or poor-quality healthcare. It was possible to extract data easily about other causes of the child’s death, such as direct culpability, low socio-economic status, problematic lifestyle, or low healthcare literacy of the mother.

Overall, the results of our study indicate the following conclusions:The proportion of forensic and medical autopsies varies significantly from one department to another (while in OL, there are 4.4 times more forensic than medical, in BR only 1.5 times more).The most significant differences between departments were found in diagnosing asphyxia and SIDS (OVA—asphyxia 0%, OL 40%; SIDS OVA 23%, OL 5%).Most SIDS are diagnosed between one month and three months, with infant mortality generally decreasing after six months. The research results confirm Meadow’s theory, and this trend has not changed over the years.It can be concluded that diagnosis suffocation is found more significant where SIDS is not diagnosed as a cause of death.The diagnosis of suffocation is typical for up to 4 months of infant life; predominantly, infants die of suffocation in the first month of age.There was a variation in the frequency of diagnosis of suffocation (BR—16.8% of total cases of the workplace, OL 9.3%, OVA 3%) throughout the study period.Most often, only health risk factors are listed in the documentation, even for diagnoses requiring social findings.For diagnoses where social risk factors are not required (infections), these are reported statistically significantly more often than other diagnoses (OVA 1:8).Diagnoses of infections in hostile factors are tracked significantly more often (especially OL and OVA). Despite the collection of this information, diagnosis of infection is usually ultimately determined as the cause of death.A significant amount of relevant information on social and host risk factors is missing (out of 204 autopsy files, 116 files were missing backup information).

The evaluation of autopsy findings is highly subjective and predominantly reflects the personal experience of the dissecting physician as well as the conventions in the departments. This inconsistency leads to poor and non-validation statistical data. It cannot be totally excluded that hostile actions of a perpetrator cause deaths diagnosed as natural. In extreme cases, an inconsistent diagnostic system may lead to crimes against infants and even not being detected. We do not claim that diagnostics at individual departments are poor, but differences between the departments suggest divergent approaches and prerequisites for the final diagnosis. Consequently, the problem of violent infant death may seem less serious since the statistics show a false ratio of natural and homicidal deaths. Society is thus misled by the statistics of the actual number of homicides. A low percentage of cases reported as homicide leads to the underestimation of its magnitude. Prevention measures can be, therefore, ineffective as the motivation to set up any prevention is lower. 

The Society of Forensic Medicine and Forensic Toxicology of the Czech Medical Association Jan Evangelist Purkyně is also of the opinion that it is necessary to develop mandatory working protocols. Furthermore, it would be ideal to institute a panel of national experts who could be consulted in cases of infant death and could help harmonize the evaluation of autopsy results.

## Figures and Tables

**Figure 1 diagnostics-12-01512-f001:**
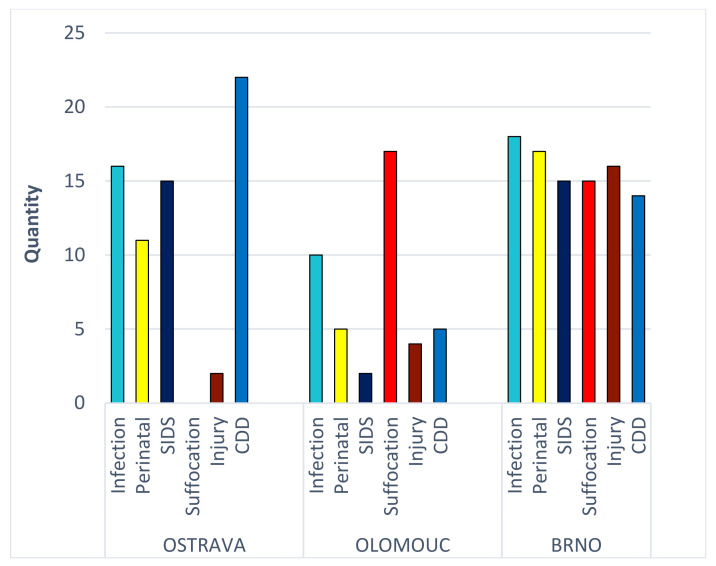
Distribution of causes of death in the three forensic departments.

**Figure 2 diagnostics-12-01512-f002:**
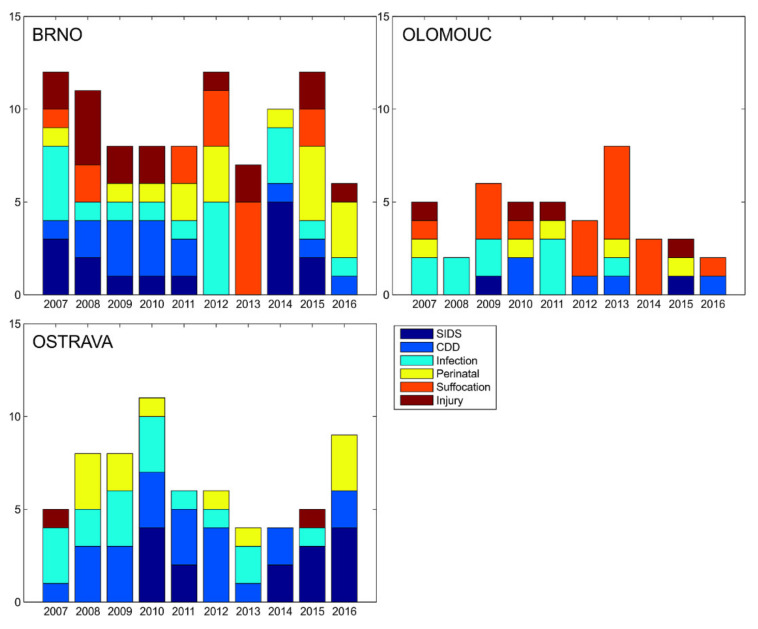
Distribution of causes of death in the three forensic departments over time.

**Figure 3 diagnostics-12-01512-f003:**
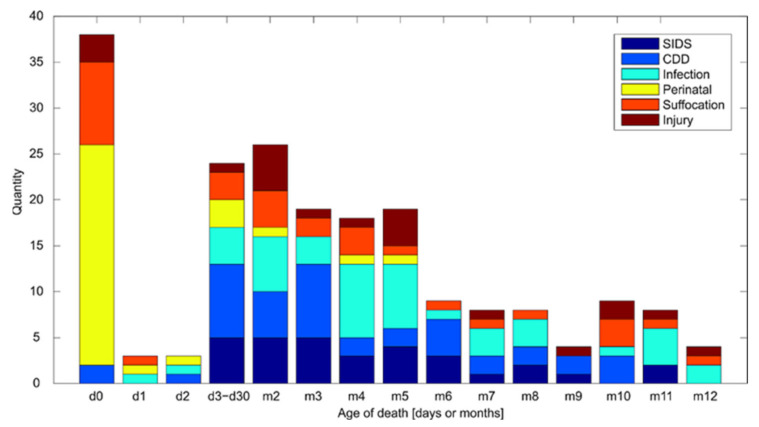
Distribution of the cause of death based on the age at the time of death. Combined data from all three departments (absolute frequency) (d—day; m—month).

**Figure 4 diagnostics-12-01512-f004:**
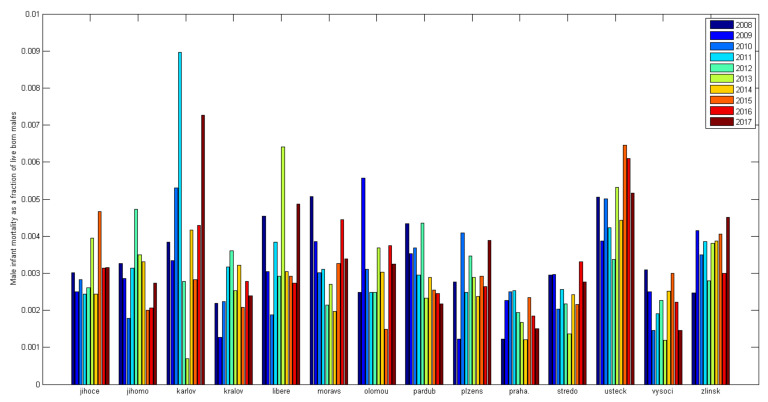
Male infant mortality in the regions (calculated from ČSÚ data [54]).

**Figure 5 diagnostics-12-01512-f005:**
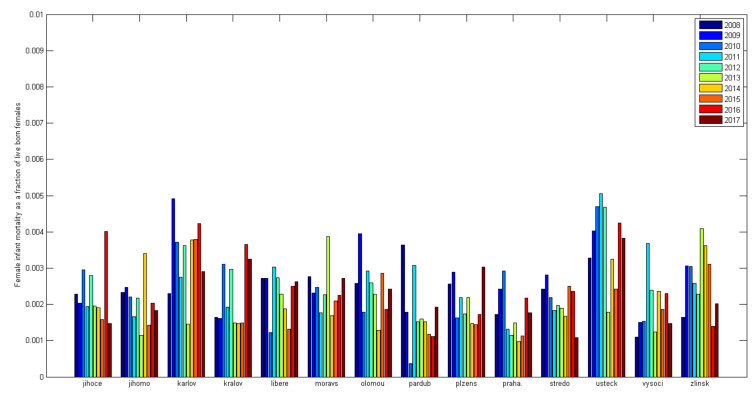
Female infant mortality in the regions (calculated from ČSÚ data [54]).

**Table 1 diagnostics-12-01512-t001:** Number and type of autopsies of infants up to 12 months old in the three departments.

	Forensic	Medical	Total	%
OV	47	19	66	32%
OL	35	8	43	21%
BR	57	38	95	47%
Total	139	65	204	100%

**Table 2 diagnostics-12-01512-t002:** Risk factors distribution (see the text for details).

	Host	Soc	Health		Absent
SIDS	0:2	0:2	2:0	OVA	13:2
CDD	0:1	0:1	1:0		21:1
Infection	4:5	1:8	5:4		7:9
Perinatal	0:3	0:3	1:2		8:3
Suffocation	0:0	0:0	0:0		0:0
Injury	0:0	0:0	0:0		2:0
SIDS	1:0	0:1	1:0	OL	1:1
CDD	3:2	0:5	2:3		0:5
Infection	6:1	1:6	3:4		3:7
Perinatal	2:1	0:3	1:2		2:3
Suffocation	7:5	6:6	10:2		5:12
Injury	0:1	0:1	1:0		3:1
SIDS	0:4	1:3	4:0	BR	11:4
CDD	1:3	0:4	1:3		10:4
Infection	1:5	2:4	3:3		12:6
Perinatal	3:4	2:5	5:2		10:7
Suffocation	9:1	6:4	9:1		5:10
Injury	11:5	9:7	13:3		3:13

Note: host—hostile behavior of the mother; soc—adverse socio-economic factors; health—adverse factors related to health; SIDS—sudden infant death syndrome; CDD—congenital developmental defects; absent—there are no relevant documentation; OVA—Ostrava; OL—Olomouc; BR—Brno.

**Table 3 diagnostics-12-01512-t003:** Number of diagnosed cases of SIDS in children older than 1 year (only those countries that showed a non-zero number listed).

GEO/TIME	2007	2008	2009	2010	2011	2012	2013	2014	2015	2016	2017	2018
European Union—27 countries	5	16	15	5	7	4	1	6	1	1	:	:
Austria	0	1	0	0	0	0	0	0	0	0	0	0
Belgium	1	1	2	1	0	0	0	0	0	0	2	0
Czechia	0	0	0	0	1	0	0	2	0	0	1	2
Finland	0	0	0	0	1	0	0	1	0	0	0	0
Germany	0	5	5	0	0	0	0	0	0	0	0	0
Hungary	0	0	1	0	0	0	0	0	0	1	0	0
Ireland	1	3	2	1	0	2	1	3	0	0	0	0
Latvia	0	0	0	0	0	1	0	0	0	0	1	0
Norway	3	3	0	0	2	3	3	1	1	1	1	1
Poland	3	3	2	0	0	0	0	0	0	0	0	0
Portugal	0	0	0	0	0	:	:	0	1	:	0	0
Spain	0	3	3	3	5	0	0	0	0	0	0	0
Sweden	0	0	0	0	0	1	0	0	0	0	0	0
Switzerland	0	0	0	1	0	0	0	0	0	0	0	0
United Kingdom	8	7	3	8	0	0	1	0	0	0	0	0

## Data Availability

Data are available on request due to restrictions, e.g., privacy or ethical reasons. The data presented in this study are available on request from the corresponding author. The data are not publicly available due to the existence of confidential data in the file.

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
