# Peer review of "Indeterminacy of the Diagnosis of Sudden Infant Death Syndrome Leading to Problems with the Validity of Data"

_diagnostics, 2022, doi:10.3390/diagnostics12071512_

Round 1

Reviewer 1 Report

We learn some important facts in your work; differences in the cause of death in different settings, inconsistency of criteria for individual causes of death as a result of different experiences and abilities, insufficient knowledge about possible homicide and poor knowledge of the social conditions in which ISD occurs. Therefore, in conclusion, it should be insisted on the harmonization of criteria in the event of sudden death, ie the creation of guidelines for a possible definition according to the rules of COR / LOE. This would reduce the possibility of authoritarianism in interpreting the cause of death.The main disadvantage of the autopsy finding is that there is no talk of additional genetic analysis related to SID (e.g. LQT interval), and this is possible. The work is otherwise excellent, and the mentioned shortcomings can also be mentioned in possible future guidelines.

I think that in the future we should organize, based on your knowledge, an initiative towards the global level for the formation of criteria for determining the death (Classical of reccomendation). It would also be good to introduce genetic research into the possible causes of AIDS in forensic testing.

Author Response

Dear Madam/Sir.

Thank you very much for your positive comments. You have captured the intentions of our text, which concern both the differences in diagnostics between different workplaces and the potential failure to detect possible crimes. We very much appreciate your revision.

Yours sincerely

Martin Dobiáš

Reviewer 2 Report

The manuscript with title “Indeterminacy of the Diagnosis of Sudden Infant Death Syndrome Leading to Problems with the Validity of Data” brings forward a very relevant and important topic.

The scope of the manuscript is of high local interest and could possibly generate further interest but presumably this might be limited due the relatively narrow framing of the problem as presented in the background. Generally the manuscript needs a bit of work to get the English language in order. A professional review of the language would benefit the manuscript quite a bit.

The abstract needs to be focused and more coherent. When you read the abstract you are lead to believe this is an article about SIDS which is not the impression given when reading the full text?

The background is interesting as such although the PUBMED search raised some thoughts, the term “misdiagonsis” seems specific and narrow. Be that as it may but there are formulated national guidelines for some European countries concerning infant death and SIDS which I think could have been an interesting and rich part to include in the background. Not the least to show that the important problem the authors bring forward has not only been considered but have also been operationalised through constructive gathering of knowledge resulting in working guidelines.

In the material and methods section the authors state that “We applied a retrospective content analysis of autopsy records of both medical and forensic autopsies of infants who died during a period of 10 years”. This seemed like an interesting approach but nothing more about this process is revealed in the manuscript. This makes it very hard to evaluate this important data collection process. How does this differ methodologically from a prospective content analysis?

In the section named measures the authors spend time introducing the following:  “The causes and mechanisms of death based on ICD10 diagnostics were grouped into the following two categories: Violent (categories 2 and 3), sudden, and unexpected (categories 2, 3, and 6) versus expected (categories 1, 4, and 5).” This whole section is not referenced again and not used in the analysis. This gives the impression the manuscript needs further work.

The statistical analyses section certainly needs work. The statement saying that “The frequency of variables was recorded in a pre-structured sheet, which was validated in a pilot study“ is actually not very clear. I can only guess what is meant by the frequency of variables? Another statement which surprises is the following: “The tested hypothesis assumes the differences in diagnostic procedures across the departments in all the monitored years.” I presume this is a kind of general statement pertaining to statistical hypotheses that should be tested but the statement needs to be qualified and detailed so it gives meaning to what is being done in this study. I would also like to see a description of statistical methods which corresponds to the various statements made in the manuscript. This is especially urgent when the reasoning is based on multiple statistical adjustment procedures, which is seen in this manuscript but is not acknowledged in this section.

The results section of this manuscript also needs a lot of work. Here we are presented with many claims that are either clearly unsubstantiated or not properly supported. It sort of becomes an exercise where we are encouraged to believe the authors claims more or less without proper proof. I will give a few examples of what I mean.

For instance the authors claim that: “Although the regions do not differ fundamentally in socio-demographic characteristics, the differences in the diagnoses of the causes of death between individual departments are significant. The differences cannot be explained by the different composition of the population nor by the socio-economic differences.” This is a bold statement and maybe even plausible but I see no evidence whatsoever in the manuscript that support this statement.

Minor comment is that the authors talk about the acronyms P1, P2 and P3 in the results based on how the authors formulate their research questions. I find that since these acronyms are only used once in the manuscript it would be better just to mention what they mean in clear text. Therefore I find them unnecessary.

“A precise test with a four-way representative table of death by suffocation in the Olomouc and Ostrava departments rejects the homogeneity hypothesis p<0.001.”                      I can probably guess how this fourfold table is composed but it would have been very good to have seen it.

Building some reasoning around the statement (which begins a new paragraph) -  “The cohort distribution throughout the years is relatively balanced and does not show any extremes.” becomes kind of meaningless. What properties are considered is not explained at all.

The statement that “Differences in causes of death rates between the years for individual departments are not statistically significant (p = 0.984).” means what? That the sample size is small or something else?

In relation to figure 2 the following text appears “The distribution of causes of death across departments shown in the time series shows that differences in cause of death diagnoses across departments do not change significantly.” This is a conclusion impossible to draw from the presented figure.

Table 2 is problematic since the elements are not properly explained. Should we guess how the authors think we should understand the semicolon? The concepts “Hostile behavior of the mother (host), adverse socio-demographic factors (soc) and adverse factors related to health (health) are combined into one indicator.” are said to be used in this table but the definitions are simply not there.

These are a few examples of problems found but these problems will spill over to the discussion and also the extensive set of conclusions made by the authors. Even though the discussion is interesting many claims are not properly founded and the same goes for the conclusions.

Author Response

Dear Madam/Sir.

We very much appreciate your review which helped us to improve our text. Thank you for your comments. We have carefully read all of your comments and recommendations. We have fully accepted all of them, taking particular care to explain all information clearly and accurately and to remove redundant and misleading information.

Yours sincerely

Martin Dobiáš

The manuscript with title “Indeterminacy of the Diagnosis of Sudden Infant Death Syndrome Leading to Problems with the Validity of Data” brings forward a very relevant and important topic.

The scope of the manuscript is of high local interest and could possibly generate further interest but presumably this might be limited due the relatively narrow framing of the problem as presented in the background. Generally the manuscript needs a bit of work to get the English language in order. A professional review of the language would benefit the manuscript quite a bit.

Thank you very much for your comments. A native speaker has checked the text; however, we realized that in places, the text is unintelligible on re-reading it. Therefore, new proofreading of the language has been carried out.

The abstract needs to be focused and more coherent. When you read the abstract you are lead to believe this is an article about SIDS which is not the impression given when reading the full text?

Fully accepted, abstract modified

The background is interesting as such although the PUBMED search raised some thoughts, the term “misdiagonsis” seems specific and narrow. Be that as it may but there are formulated national guidelines for some European countries concerning infant death and SIDS which I think could have been an interesting and rich part to include in the background. Not the least to show that the important problem the authors bring forward has not only been considered but have also been operationalised through constructive gathering of knowledge resulting in working guidelines.

Fully accepted

The information was added

In the material and methods section the authors state that “We applied a retrospective content analysis of autopsy records of both medical and forensic autopsies of infants who died during a period of 10 years”. This seemed like an interesting approach but nothing more about this process is revealed in the manuscript. This makes it very hard to evaluate this important data collection process. How does this differ methodologically from a prospective content analysis?

Fully accepted

The information was added

In the section named measures the authors spend time introducing the following:  “The causes and mechanisms of death based on ICD10 diagnostics were grouped into the following two categories: Violent (categories 2 and 3), sudden, and unexpected (categories 2, 3, and 6) versus expected (categories 1, 4, and 5).” This whole section is not referenced again and not used in the analysis. This gives the impression the manuscript needs further work.

Fully accepted

Thank you for your comment.

This section was redundant

The statistical analyses section certainly needs work. The statement saying that “The frequency of variables was recorded in a pre-structured sheet, which was validated in a pilot study“ is actually not very clear. I can only guess what is meant by the frequency of variables? Another statement which surprises is the following: “The tested hypothesis assumes the differences in diagnostic procedures across the departments in all the monitored years.” I presume this is a kind of general statement pertaining to statistical hypotheses that should be tested but the statement needs to be qualified and detailed so it gives meaning to what is being done in this study. I would also like to see a description of statistical methods which corresponds to the various statements made in the manuscript. This is especially urgent when the reasoning is based on multiple statistical adjustment procedures, which is seen in this manuscript but is not acknowledged in this section.

Fully accepted

Thank you very much. We have taken this comment very seriously. We have read all the descriptions of the methods thoroughly and have tried to describe the methods better and in more detail.

The results section of this manuscript also needs a lot of work. Here we are presented with many claims that are either clearly unsubstantiated or not properly supported. It sort of becomes an exercise where we are encouraged to believe the authors claims more or less without proper proof. I will give a few examples of what I mean.

For instance the authors claim that: “Although the regions do not differ fundamentally in socio-demographic characteristics, the differences in the diagnoses of the causes of death between individual departments are significant. The differences cannot be explained by the different composition of the population nor by the socio-economic differences.” This is a bold statement and maybe even plausible but I see no evidence whatsoever in the manuscript that support this statement.

Fully accepted

Based on the description of the methods, we also modified the wording of the results to make them easier to understand.

                Minor comment is that the authors talk about the acronyms P1, P2 and P3 in the results based on how the authors formulate their research questions. I find that since these acronyms are only used once in the manuscript it would be better just to mention what they mean in clear text. Therefore I find them unnecessary.

Fully accepted

The acronyms were removed

“A precise test with a four-way representative table of death by suffocation in the Olomouc and Ostrava departments rejects the homogeneity hypothesis p<0.001.”        I can probably guess how this fourfold table is composed but it would have been very good to have seen it.

Fully accepted

Thank you very much for your comment. This part of the text was unintelligible and has been reworded to be more precise.

Building some reasoning around the statement (which begins a new paragraph) -  “The cohort distribution throughout the years is relatively balanced and does not show any extremes.” becomes kind of meaningless. What properties are considered is not explained at all.

Fully accepted

Thank you very much for your comment. This part of the text was unintelligible and has been reworded to be more precise.

The statement that “Differences in causes of death rates between the years for individual departments are not statistically significant (p = 0.984).” means what? That the sample size is small or something else?

Fully accepted

Thank you very much for your comment. This part of the text was unintelligible and has been reworded to be more precise.

In relation to figure 2 the following text appears “The distribution of causes of death across departments shown in the time series shows that differences in cause of death diagnoses across departments do not change significantly.” This is a conclusion impossible to draw from the presented figure.

Fully accepted

Thank you very much for your comment. This part of the text was unintelligible and has been reworded to be more precise.

Table 2 is problematic since the elements are not properly explained. Should we guess how the authors think we should understand the semicolon? The concepts “Hostile behavior of the mother (host), adverse socio-demographic factors (soc) and adverse factors related to health (health) are combined into one indicator.” are said to be used in this table but the definitions are simply not there.

Fully accepted

Thank you very much for your comment. This part of the text was unintelligible and has been reworded to be more precise.

These are a few examples of problems found but these problems will spill over to the discussion and also the extensive set of conclusions made by the authors. Even though the discussion is interesting many claims are not properly founded and the same goes for the conclusions.

Thank you very much indeed for all your comments. We believe that their feedback has helped us significantly improve the text's quality.

Round 2

Reviewer 2 Report

As stated in my previous review of this manuscript with title “Indeterminacy of the Diagnosis of Sudden Infant Death Syndrome Leading to Problems with the Validity of Data” brings forward a very relevant and important topic.

In this iteration the authors have remedied many issues found in the previous iteration of the manuscript. I still believe that the scope of the manuscript is of high local interest and could possibly generate further interest but presumably this might be limited due the relatively narrow framing of the problem as presented in the background. The English language has improved considerably although there still are things that could be sorted out. I am convinced another iteration of language checking will yield a presentable manuscript. My general impression is that after sorting out some further issues, the manuscript could be suitable for publishing.

The literature search is better presented although I still consider omitting  relevant official national guidelines on how to approach the issue of SIDS a deficiency in this literature review. I know the work is already done and now is not the time to change but for future research, this potentially rich source of information should perhaps not be omitted.

The statistical analyses section still needs some work. I will repeat my question from the last time: The statement saying that “The frequency of variables was recorded in a pre-structured sheet, which was validated in a pilot study“ is actually still not very clear. I can only guess what is meant by the frequency of variables? Are you referring to the frequency of values for the included variables?

I am also still lacking a presentation for when the reasoning is based on multiple statistical adjustment procedures, which is seen in this manuscript but is not acknowledged in the statistical analyses section. For example a statement like: “Although the regions do not differ fundamentally in socio-demographic characteristics, the differences in the composition of causes of death among the three departments are profound”. How do you know this without reasonable statistical adjustment for several variables? If these analyses are not performed by the authors perhaps some suitable reference is at hand?

 Aims and research questions need work:

First statement is: This study explored different diagnostic procedures used to determine the cause of Sudden Infant Death Syndrome across forensic settings

The addition to the sentence “.. and consequently the issue of reduced validity of data relating to violence against children. “ is as stated a possible consequence and not a hypothesis

Next statement is also problematic: “The following premises were derived from the research questions: “ What research questions? They are never presented as far as  I can see?

Are they perhaps hidden in the statements below? Maybe better to make them into proper research questions.

- Differences between departments regarding the diagnosed causes of death.

- Differences between departments regarding how information is collected on social risk factors.

- Differences between departments regarding the identification of social factors.

In the measures section the following appear:

What specific differences exist in the number of causes of death in children between the clinics?

If the differences between clinics are significant, can the data on the causes of death of children be considered valid?

Could the low validity of data reported in court files indicate a bias in the extent of violence against these children?

For the sake of clarity I would now, since the manuscript have improved a lot, recommend the authors to collect all the research questions and hypotheses in one place and not, as it is now, surprise is with more research questions in the measures section. I realize the appeal but still will advice against this format. Formulate clear and concise hypotheses and research questions. Limit the number of them to what is important. I also urge the authors to limit the conclusions to what really matters which should be possible answers to precisely formulated research questions. An overall short conclusion summarizing what’s the important takeaway could also be good.

I also reacted on a statement on lines 735-736 saying: “In all high-income countries, SIDS currently represents the largest group in the mortality rate of 1-12-month old infants.” Is this really true? I have seen number on infant deaths from the US 2019 stating something else. There the claim was that SIDS only came in third after birth defects and premature/low birth weight cases.

Minor comments:

That the data comprises infant deaths from 2007 to 2016 is only mentioned explicitly in the abstract and also in parenthesis?

In the abstract the statement that: “The most significant differences in the diagnosis of Sudden Infant Death Syndrome and suffocation could be observed in the group of sudden and unexpected death.” is unclear since over what entity the comparison is made is never mentioned.

In the introduction two rather important blocks of facts are put inside parentheses for reasons unclear to me.

Line 79, “diagnostic parameters” ? For instance a diagnostic criteria I can understand, not sure what a parameter might be? For example, a standard deviation is a parameter.

Example of a language issue: line 97 – “was” should be “were”

Line 166: Stress factor? What kind of stress is this? Surely another word could be found?

Generally chapter division is not logical: 1 Introduction, 2 Literature Review, 2 Materials and Methods…?

Language examples: (there are further issues here and there)

Line 464: “The highest fraction of autopsies” I suggest - proportion of ….

Line 480 “If we concentrate on a single..” I suggest  - If we focus on …

Lines 650-651: To interpret these findings, it has to must be admitted that each department has different diagnostic procedures in place…   Very unclear with phrase “it has to must..”

Author Response

The literature search is better presented although I still consider omitting  relevant official national guidelines on how to approach the issue of SIDS a deficiency in this literature review. I know the work is already done and now is not the time to change but for future research, this potentially rich source of information should perhaps not be omitted.

Thank for the recognition.

There is a problem with the official national guidelines for SIDS in European countries. In the Czech Republic and neighboring countries, at least according to our information, there are none (except Italy). They all work mainly according to the information contained in textbooks.

This information has been incorporated into the text.

The statistical analyses section still needs some work. I will repeat my question from the last time: The statement saying that “The frequency of variables was recorded in a pre-structured sheet, which was validated in a pilot study“ is actually still not very clear. I can only guess what is meant by the frequency of variables? Are you referring to the frequency of values for the included variables?

We accept the comment.

We have tried to clarify the sentence.

I am also still lacking a presentation for when the reasoning is based on multiple statistical adjustment procedures, which is seen in this manuscript but is not acknowledged in the statistical analyses section. For example a statement like: “Although the regions do not differ fundamentally in socio-demographic characteristics, the differences in the composition of causes of death among the three departments are profound”. How do you know this without reasonable statistical adjustment for several variables? If these analyses are not performed by the authors perhaps some suitable reference is at hand?

We apologize for this error.

The information was mistakenly left in the results section.

Thank you for your attention.

It belongs in the discussion where it has been moved and has been supported by statistical evidence.

First statement is: This study explored different diagnostic procedures used to determine the cause of Sudden Infant Death Syndrome across forensic settings

The addition to the sentence “.. and consequently the issue of reduced validity of data relating to violence against children. “ is as stated a possible consequence and not a hypothesis

Next statement is also problematic: “The following premises were derived from the research questions: “ What research questions? They are never presented as far as  I can see?

Are they perhaps hidden in the statements below? Maybe better to make them into proper research questions.

- Differences between departments regarding the diagnosed causes of death.

- Differences between departments regarding how information is collected on social risk factors.

- Differences between departments regarding the identification of social factors.

In the measures section the following appear:

What specific differences exist in the number of causes of death in children between the clinics?

If the differences between clinics are significant, can the data on the causes of death of children be considered valid?

Could the low validity of data reported in court files indicate a bias in the extent of violence against these children?

Thank you for your comments.

They have been fully accepted.

For the sake of clarity I would now, since the manuscript have improved a lot, recommend the authors to collect all the research questions and hypotheses in one place and not, as it is now, surprise is with more research questions in the measures section. I realize the appeal but still will advice against this format. Formulate clear and concise hypotheses and research questions. Limit the number of them to what is important. I also urge the authors to limit the conclusions to what really matters which should be possible answers to precisely formulated research questions. An overall short conclusion summarizing what’s the important takeaway could also be good.

Thank you for your comments.

They have been fully accepted.

I also reacted on a statement on lines 735-736 saying: “In all high-income countries, SIDS currently represents the largest group in the mortality rate of 1-12-month old infants.” Is this really true? I have seen number on infant deaths from the US 2019 stating something else. There the claim was that SIDS only came in third after birth defects and premature/low birth weight cases.

Thank you for your thoughtfulness.

The sentence has been rephrased to make the statement correct.

That the data comprises infant deaths from 2007 to 2016 is only mentioned explicitly in the abstract and also in parenthesis?

The bracket has been removed from the abstract. The information is also included in the Sample section, line 200.

In the abstract the statement that: “The most significant differences in the diagnosis of Sudden Infant Death Syndrome and suffocation could be observed in the group of sudden and unexpected death.” is unclear since over what entity the comparison is made is never mentioned.

Fully accepted.

In the introduction two rather important blocks of facts are put inside parentheses for reasons unclear to me.

The bracket has been removed.

Line 79, “diagnostic parameters” ? For instance a diagnostic criteria I can understand, not sure what a parameter might be? For example, a standard deviation is a parameter.

Fully accepted.

Example of a language issue: line 97 – “was” should be “were”

Fully accepted.

Line 166: Stress factor? What kind of stress is this? Surely another word could be found?

Fully accepted. Information has been added.

Generally chapter division is not logical: 1 Introduction, 2 Literature Review, 2 Materials and Methods…?

It was corrected.

Language examples: (there are further issues here and there)

Line 464: “The highest fraction of autopsies” I suggest - proportion of ….

Line 480 “If we concentrate on a single..” I suggest  - If we focus on …

Lines 650-651: To interpret these findings, it has to must be admitted that each department has different diagnostic procedures in place…   Very unclear with phrase “it has to must..”

Thank you for your careful reading.

All comments have been accepted.
